# Dietary Habits, Residential Air Pollution, and Chronic Obstructive Pulmonary Disease

**DOI:** 10.3390/nu17122029

**Published:** 2025-06-18

**Authors:** Dong Liu, Junyi Ma, Xia-Lin Cui, Yunnan Zhang, Tong Liu, Li-Hua Chen

**Affiliations:** 1Department of Nutrition and Food Hygiene, School of Public Health, Nantong University, Nantong 226019, China; dongliusph@ntu.edu.cn (D.L.); zyn1767450173@163.com (Y.Z.); 2Institute for Applied Research in Public Health, School of Public Health, Nantong University, Nantong 226019, China; 3Institute of Pain Medicine and Special Environmental Medicine, Nantong University, Nantong 226019, China; mjy070707@163.com (J.M.); 17660570506@163.com (X.-L.C.); 4Department of Nutrition and Food Hygiene, School of Public Health, Suzhou Medical College, Soochow University, Suzhou 215123, China

**Keywords:** chronic obstructive pulmonary disease, dietary patterns, particulate matter, nitrogen oxides

## Abstract

**Background:** The role of dietary patterns in the development of chronic obstructive pulmonary disease (COPD), particularly under varying levels of ambient air pollution, remains insufficiently understood. **Aims:** We aimed to investigate the association between adherence to multiple established dietary patterns and the risk of incident COPD, and to assess potential effect modification by exposure to ambient air pollutants. **Methods:** We conducted a prospective study including 206,463 participants from the UK Biobank free of COPD at baseline. Individual-level residential air pollution exposure was estimated for the year 2010. Nine dietary indices were derived from 24 h dietary recalls. Associations with incident COPD were assessed using Cox proportional hazards models. Effect modification was examined using smoking-specific tertiles of nitrogen oxides (NO, NO_2_, and NO_x_) and particulate matter (PM_2.5_, PM_2.5–10_, and PM_10_). **Results:** Greater adherence to healthy dietary patterns was associated with a 14% to 34% reduced risk of COPD (highest vs. lowest quintile). In contrast, high adherence to the Unhealthful plant-based diet index (PDI) was associated with a 34% increased risk (HR = 1.34, 95% CI: 1.16–1.54). Notably, the protective associations of the AHA, EAT-Lancet, and MIND dietary patterns were most pronounced in settings with relatively high air pollution, as evidenced by elevated levels in at least four air quality indicators (*p* for interaction < 0.05). **Conclusions:** Adherence to AHA, EAT-Lancet, and MIND dietary patterns is associated with a reduced risk of incident COPD, with potentially amplified benefits observed in areas with higher ambient air pollution.

## 1. Introduction

Chronic obstructive pulmonary disease (COPD), a progressive respiratory condition characterized by persistent airflow limitation, remains a leading contributor to global morbidity and mortality [1]. While cigarette smoking and long-term exposure to ambient air pollutants are well-established primary risk factors [2], emerging evidence highlights the potential role of other modifiable lifestyle factors—particularly diet—in modifying the risk and progression of this disease [2,3,4].

Previous studies have primarily focused on the effects of individual nutrients or food groups on respiratory health. For instance, higher intakes of antioxidants such as vitamins [5,6], phytochemicals [7], dietary fiber [8,9], and polyunsaturated fatty acids [10] have been associated with improved lung function or reduced risk of respiratory conditions. However, these reductionist approaches may not fully capture the complexity of dietary exposures and their cumulative effects on chronic disease risk. Consequently, dietary pattern analysis, which considers the combined impact of multiple foods and nutrients, has emerged as a valuable approach in nutritional epidemiology [2,3,4,11]. Several well-established dietary indices—such as the Alternative Healthy Eating Index-2010 (AHEI-2010) [4] and the alternate Mediterranean diet (AMED) score [11]—have been linked to reduced risk of COPD, yet evidence regarding their associations with COPD remains inconsistent and relatively understudied [12,13,14,15,16,17,18,19].

In parallel, exposure to air pollutants—particularly nitrogen oxides (NOx) and particulate matter (PM)—is a major environmental determinant of COPD onset and progression [2]. Fine (PM_2.5_) and coarse (PM_2.5–10_) particles can penetrate deep into the respiratory tract and trigger oxidative stress and systemic inflammation, key pathophysiological mechanisms in COPD development [20]. Notably, these biological pathways may be modified by diet; for example, diets rich in anti-inflammatory and antioxidant compounds might counteract the adverse effects of air pollution on lung health [21,22,23,24,25]. Despite this biological plausibility, few studies have evaluated whether dietary patterns could attenuate the detrimental impact of long-term air pollution exposure on COPD risk.

Most existing research has examined dietary patterns and air pollution exposures in isolation, leaving a critical gap in our understanding of their potential interplay in relation to COPD. Moreover, large-scale prospective evidence evaluating these associations across varying levels of environmental exposure is scarce. To address these research gaps, the present study leverages data from the UK Biobank—a large prospective cohort with extensive dietary and high-resolution geospatial air pollution data—to (1) examine the prospective associations between adherence to several well-established dietary patterns and the incidence of COPD, and (2) evaluate whether long-term exposure to ambient NOx or PM modifies these associations.

## 2. Material and Methods

### 2.1. Study Design and Population

This study utilized data from the UK Biobank, a large-scale, population-based study that recruited 502,493 participants aged 40–69 years between 2006 and 2010. The UK Biobank collected extensive baseline data on sociodemographic characteristics, lifestyle behaviors, physical measurements, biological samples, and health outcomes [26]. Participants were excluded if they had missing or incomplete 24 h dietary recall data, implausible total energy intake, or a diagnosis of COPD at baseline. After exclusions, 206,463 participants were included in the final analytical sample (Figure 1). All procedures were conducted in accordance with the Declaration of Helsinki.

### 2.2. Dietary Assessment and Derivation of Dietary Pattern Scores

Between April 2009 and June 2012, dietary intake was assessed using a validated online 24 h dietary recall tool developed specifically for large population-based studies [27]. A subset of participants (61%, *n* = 125,319) completed repeated 24 h dietary recalls during follow-up in the present study: 47,403 participants completed two assessments, 42,013 completed three, 30,192 completed four, and 5711 completed five. For participants with multiple assessments, the average dietary intake across all available recalls was used to derive dietary pattern scores. The food group components, corresponding food items, Field-IDs, questionnaire units, and defined portion sizes, along with the codes from the Composition of Foods Integrated Dataset (CoFID) applied in this study, are detailed in Appendix A. As shown in Appendix A, data on nutrient, total energy, and alcohol intake parameters were derived directly from the UK Biobank, calculated using the CoFID, or derived from other data repositories.

Energy and nutrient intakes were adjusted for total energy intake using the residual method to reduce confounding by total energy intake and to better reflect dietary composition independent of caloric consumption [28,29]. Nine dietary pattern scores were constructed based on predefined indices: the American Heart Association (AHA) diet score, the AMED, AHEI-2010, Dietary Approaches to Stop Hypertension (DASH), the EAT-Lancet diet, the Mediterranean-DASH Intervention for Neurodegenerative Delay (MIND) diet, and three plant-based diet indices (the Overall, Healthful, and Unhealthful PDIs). Detailed scoring algorithms are described in Appendix A and summarized in Appendix A. A systematic summary of the individual components comprising these dietary patterns is further presented in Appendix A. For analytical purposes, dietary pattern scores were categorized into sex-specific quintiles. Standardized Z-scores were also calculated to facilitate comparison across dietary patterns and to allow continuous modeling of associations with outcomes [30].

### 2.3. Ascertainment of Outcomes

Participants were followed from baseline until the first occurrence of an outcome event, death, or 30 September 2020, whichever came first. The primary outcome was incident COPD, defined using a combination of self-reported diagnoses, linked hospital inpatient records, and primary care data. Diagnoses were identified according to the International Classification of Diseases (ICD-9 codes: 491.2, 496; ICD-10 code: J44). Secondary outcomes included COPD-related mortality, determined from national death registry records that captured both primary and contributory causes of death coded under ICD-10 (Appendix A).

### 2.4. Covariates

Baseline covariates were selected a priori based on known or plausible associations with COPD risk and dietary exposures. These included demographics, lifestyle factors, clinical factors, and occupational exposures. Demographics comprised age (continuous), sex (male/female), and self-reported ethnicity (White or non-White); socioeconomic status based on the Townsend deprivation index, a standardized composite score based on national census data, with higher values indicating greater deprivation [31]; and educational attainment, categorized as high (college/university degree), middle (A/AS levels or O levels/GCSEs or equivalent), or low (no formal qualifications). Lifestyle factors comprised body mass index (BMI), calculated as weight in kilograms divided by height in meters squared (kg/m^2^); physical activity, categorized as low, moderate, or high based on Metabolic Equivalent Task (MET)-minutes/week scores derived from validated questionnaires [32]; smoking status, classified as never, previous, or current; alcohol consumption, categorized as never, previous, or current; and total energy intake, calculated in kilocalories per day and modeled as a continuous variable. Clinical factors comprised a self-reported or medically recorded history of cardiovascular disease (CVD), type 2 diabetes mellitus (T2DM), hypertension, or respiratory diseases other than COPD (e.g., asthma, emphysema, cystic fibrosis, pulmonary tuberculosis). Hypertension was further defined by recorded systolic/diastolic blood pressure (≥140/90 mmHg) or current use of antihypertensive medications. Occupational exposures comprised a binary variable capturing the presence or history of occupation-related respiratory conditions or exposure to harmful agents. Estimates of individual-level exposure to ambient air pollution were obtained from the UK Biobank air quality database, based on land-use regression models developed under the ESCAPE project [33]. These models provide annual average concentrations of nitrogen oxide (NO), nitrogen dioxide (NO_2_), and PM with aerodynamic diameters <2.5 μm (PM_2.5_), between 2.5 and 10 μm (PM_2.5–10_), and <10 μm (PM_10_) at each participant’s residential address in 2010. NO_x_ was computed as the sum of NO and NO_2_ concentrations. Baseline lung function was assessed in approximately 81% of participants using a handheld pneumotachograph spirometer (Pneumotrac 6800, Vitalograph Ltd., Buckingham, UK) which measured forced expiratory volume in one second (FEV_1_) and forced vital capacity (FVC) [34]. The FEV_1_/FVC ratio was also calculated. The procedure included checks for contraindications (e.g., recent chest infection, recent surgery, pregnancy), and participants reporting any were excluded from testing. Furthermore, it was recorded whether participants had used an inhaler within the last hour.

### 2.5. Statistical Analyses

Continuous variables are described using medians and interquartile ranges (IQRs), while categorical variables are presented as frequencies and percentages. Cox proportional hazards models were used to estimate hazard ratios (HRs) and 95% confidence intervals (CIs). Three models were constructed: Model 1: unadjusted; Model 2: adjusted for age, sex, and ethnicity; Model 3: further adjusted for Townsend index, education, BMI, physical activity, smoking, alcohol, total energy intake, CVD, T2DM, hypertension, other respiratory diseases, occupation-related respiratory risks, and air pollutants. Non-linear associations were examined using restricted cubic spline models.

Four sensitivity analyses were performed: (1) excluded incident COPD cases within two years; (2) was restricted to participants with ≥2 valid 24 h dietary recalls; (3) excluded baseline respiratory diseases/occupational breathing issues; (4) included only those with baseline spirometry-confirmed normal lung function [2] (FEV1/FVC >0.70). Stratified analyses were used to assess effect modification by prespecified variables, testing multiplicative interactions via product terms in regression models. Smoking-specific tertiles of air pollution exposure were used to minimize confounding. The predictive value of dietary patterns was evaluated using integrated discrimination improvement (IDI), continuous net reclassification improvement (NRI), and changes in the area under the receiver operating characteristic curve (ΔAUC). Analyses were performed using SAS version 9.4 (SAS Institute Inc., Cary, NC, USA), and visualizations were generated using R version 4.3.1 (R Foundation for Statistical Computing, Vienna, Austria).

## 3. Results

### 3.1. Characteristics of the Study Population

Table 1 and Appendix A summarize the demographic and clinical characteristics of the study population (*n* = 206,463). The cohort had a median age of 57 years (IQR: 50–63), was predominantly White (95.54%), and included 44.81% male participants. Notably, individuals in higher FEV_1_/FVC ratio quartiles tended to be younger, were more likely to be female, and had healthier profiles overall—including lower rates of current smoking, reduced total energy intake, fewer cardiovascular comorbidities (e.g., CVD and hypertension), fewer respiratory disease diagnoses, and less frequent inhaler use within the hour prior to spirometry testing.

Appendix A further present characteristics across quintiles of dietary pattern scores, comparing the lowest (Q1) and highest (Q5) quintiles. Participants in the highest quintiles of healthier dietary patterns were more likely to be older, have a lower BMI, smoke less, engage in more physical activity, attain higher education levels, consume less alcohol, and exhibit a lower prevalence of T2DM. Conversely, those in the highest quintile of the Unhealthful PDI tended to be younger, have a higher BMI, smoke more, engage in less physical activity, and have lower education levels. Additionally, healthier dietary patterns were associated with lower deprivation levels and slightly reduced exposure to air pollution.

### 3.2. Multiple Dietary Patterns Associated with the Risk of Total COPD and COPD-Caused Mortality

A total of 2450 incident COPD cases were identified during follow-up. The vast majority of cases were ascertained through linked clinical records (hospital inpatient or death registry), with only a negligible number (*n* = 5) based solely on self-reported diagnoses without supporting clinical documentation. As shown in Table 2, compared with participants in the lowest quintile (Q1) of dietary pattern scores, those in the highest quintile (Q5) exhibited significantly lower risks of total COPD, with adjusted HRs as follows: 0.67 (95% CI: 0.58–0.77) for the AHA diet, 0.66 (95% CI: 0.58–0.76) for the AMED, 0.70 (95% CI: 0.61–0.79) for the AHEI-2010, 0.69 (95% CI: 0.61–0.79) for DASH, 0.68 (95% CI: 0.60–0.79) for EAT-Lancet, and 0.72 (95% CI: 0.63–0.82) for the MIND. Similarly, the Healthful PDI was associated with reduced risk (HR: 0.82, 95% CI: 0.72–0.94), whereas the Unhealthful PDI showed an increased risk (HR: 1.34, 95% CI: 1.16–1.54).

For COPD-related mortality (Appendix A), similar protective associations were observed for healthier dietary patterns, with adjusted HRs of 0.61 (95% CI: 0.40–0.93) for the AHA diet, 0.37 (95% CI: 0.24–0.58) for the AMED, 0.54 (95% CI: 0.36–0.80) for the AHEI-2010, 0.48 (95% CI: 0.31–0.74) for DASH, 0.55 (95% CI: 0.37–0.83) for EAT-Lancet, and 0.49 (95% CI: 0.31–0.78) for the MIND. In contrast, the Unhealthful PDI was associated with significantly higher mortality risk (HR: 1.85, 95% CI: 1.17–2.93). No significant associations were found for the Overall PDI or Healthful PDI in the mortality analysis. Similar analyses using continuous Z-scores yielded results consistent with the quintile-based approach.

### 3.3. Potential Non-Linear Associations of Multiple Dietary Patterns with Total COPD and COPD-Caused Mortality

Figure 2 and Appendix A present multivariable-adjusted restricted cubic spline analyses revealing significant linear associations for general healthy dietary patterns with both risks of COPD (all *p* values for linearity < 0.001) and COPD-related mortality (all *p* values for linearity < 0.05). The Unhealthful PDI showed an inverse linear relationship with these outcomes. Notably, non-linear trends were observed for Overall PDI and Healthful PDI scores in relation to COPD risk, as well as for the EAT-Lancet score in association with COPD mortality (all *p* values for non-linearity < 0.05).

### 3.4. Predictive Performance of Multiple Dietary Patterns in Assessing COPD Risk

As shown in Appendix A, the incorporation of different dietary patterns demonstrated variable improvements in predictive performance. The AHA diet score showed the most substantial improvement in model performance, with a statistically significant NRI of 11.9% (95% CI: 7.91–15.9; *p* < 0.001) and the highest IDI improvement of 0.09% (95% CI: 0.05–0.13; *p* < 0.001). Notably, the DASH score showed the greatest AUC improvement (ΔAUC = 0.09%, 95% CI: 0.02–0.15; *p* = 0.009). With the exception of the Overall PDI, all other dietary patterns demonstrated statistically significant enhancements in predictive performance. The NRIs ranged from 7.61% to 18.9%, while IDI values increased between 0.02% and 0.07%. Corresponding ΔAUC values varied from 0.061% to 0.084%, all reaching statistical significance (*p* < 0.05 for all comparisons).

### 3.5. Sensitivity Analyses

The results of multiple sensitivity analyses are presented in Appendix A. Our primary findings—demonstrating significant associations between healthy dietary patterns and COPD risk—remained robust across all sensitivity analyses, except for in the case of the Overall PDI score, which did not show consistent significance. Specifically, the associations persisted even after excluding participants who developed COPD within the first two years of follow-up, minimizing potential reverse causality (Sensitivity Analysis 1). The robustness of these associations was further confirmed when restricting analyses to participants with at least two valid 24 h dietary records, reinforcing the reliability of dietary exposure assessment, as shown in Sensitivity Analysis 2. Furthermore, the primary findings remained statistically unchanged when excluding participants with pre-existing respiratory diseases or occupation-related breathing problems at baseline (Sensitivity Analysis 3) and, more importantly, when restricting analyses to participants with an FEV_1_/FVC ratio > 0.70, ensuring COPD diagnoses were not confounded by restrictive lung disease (Sensitivity Analysis 4).

### 3.6. Subgroup Analyses

As shown in Appendix A, the associations between dietary pattern scores and COPD incidence remained largely consistent across all stratified subgroups, including age, sex, race, BMI, smoking status, alcohol consumption, physical activity, and comorbidities. These findings were generally similar to those observed in the overall population analysis. Figure 3 presents stratified analyses of the associations between dietary patterns and COPD risk across smoking-specific tertiles of air pollution exposure. Overall, for all dietary indices, participants in the highest tertile (T3) of pollution environments showed reduced COPD risk compared to those in the lowest tertile (T1). Notably, compared to the moderate exposures (the second tertile, T2), the apparent protective associations were most pronounced for the AHA diet in poor air condition (T3) for NO (*p* for interaction = 0.013), NO_2_ (*p* for interaction = 0.009), NO_X_ (*p* for interaction = 0.002), and PM_2.5–10_ (*p* for interaction = 0.026); the AHEI-2010 for PM_2.5–10_ (*p* for interaction = 0.043); DASH for PM_2.5–10_ (*p* for interaction <0.001); EAT-Lancet for NO (*p* for interaction = 0.042), NO_2_ (*p* for interaction = 0.002), NO_X_ (*p* for interaction = 0.011), and PM_2.5–10_ (*p* for interaction = 0.040); and MIND for NO (*p* for interaction = 0.011), NO_X_ (*p* for interaction = 0.045), PM_2.5_ (*p* for interaction = 0.013), and PM_2.5–10_ (*p* for interaction = 0.002). In addition, the risk associations of the Healthful PDI were significantly weaker in the lowest pollution tertiles for PM_2.5_ (*p* for interaction = 0.033) and PM_10_ (*p* for interaction < 0.001). 

## 4. Discussion

We observed that greater adherence to healthy dietary guidelines may reduce the incidence of COPD, while the Unhealthful PDI increases risk. Notably, these associations tend to be modified by ambient air pollution exposure levels, with the strongest protective effects observed in individuals residing in areas with higher concentrations of nitrogen oxides and PM. Our findings suggest a potential role for diet in mitigating the detrimental pulmonary effects of air pollution—a novel and timely insight given the global burden of air quality deterioration.

Our findings align with and extend the prior literature suggesting that antioxidant-rich, anti-inflammatory diets may play a protective role in respiratory health. For example, the “prudent” dietary pattern, characterized by high intakes of fruits, vegetables, whole grains, and fish, has previously been associated with greater lung function and lower COPD risk [35]. Similarly, adherence to the AHEI-2010 has been linked to favorable respiratory outcomes [4]. More recently, a “lung-healthy” diet high in green leafy vegetables and tea and low in processed meats was shown to correlate with improved respiratory parameters [3]. Our study builds upon this evidence by incorporating a broader array of dietary indices, including AHA, EAT-Lancet, MIND, and PDI scores, which were previously underexplored in relation to COPD onset.

Although the inverse associations between healthy dietary patterns and COPD risk are biologically plausible—primarily due to their potential anti-inflammatory and antioxidant effects [36]—our study lacked biomarker data (e.g., CRP, IL-6, oxidative stress markers) or intermediate phenotypes to directly substantiate these mechanistic pathways. Therefore, further longitudinal and mechanistic studies incorporating such data are warranted. Diets rich in polyphenols, vitamins C and E, fiber, and omega-3 fatty acids are known to modulate inflammatory cascades, improve endothelial function, and enhance host antioxidant defenses. These pathways may be especially relevant in populations exposed to chronic environmental pollutants, which have been shown to induce oxidative lung damage, airway remodeling, and impaired immune responses [37]. Our analyses further revealed significant interactions between dietary patterns and air pollution exposure. Individuals in the highest tertile of pollution exposure derived the greatest benefit from healthful diets. For instance, the protective associations of the AHA diet, EAT−Lancet, and the MIND were significantly stronger among those that lived in elevated NO_x_ and PM_2.5–10_ conditions. These findings underscore the potential of dietary interventions to mitigate pollution-induced respiratory damage and highlight the role of environmental context in shaping dietary effects.

While most healthy dietary patterns were inversely associated with COPD risk, the Overall PDI showed inconsistent associations. This may be due to its construction, which assigns equal positive scores to all plant foods regardless of nutritional quality, including refined grains, sugar-sweetened beverages, and other less healthful items. Similarly, although the Healthful PDI emphasizes nutritious plant-based foods, it also penalizes all animal-based foods—including fish and dairy—which may have beneficial effects on respiratory health [38]. These scoring characteristics may partly explain the attenuated associations observed. Furthermore, the Healthful PDI also demonstrated weaker associations in participants exposed to lower levels of air pollution. This could reflect a lower baseline risk of COPD, a reduced oxidative burden, and a potential ceiling effect of dietary benefits in cleaner environments, where the protective role of diet may be less pronounced.

To our knowledge, this is the first study to systematically evaluate the interplay between a wide range of well-validated dietary patterns and COPD risk while concurrently examining modification by environmental exposures. The public health relevance of our findings is particularly notable in light of the global burden of air pollution. Fine particulate matter (PM_2.5_) and nitrogen oxides (NO_x_) remain major contributors to respiratory morbidity, with no evidence of a safe exposure threshold. Smaller-sized particles penetrate deep into the pulmonary alveoli, triggering immune activation, oxidative stress, and chronic inflammation [39]. Our findings suggest that diet may serve as a modifiable behavioral factor to counteract these harmful effects, particularly in high-exposure regions.

This study has several strengths: It leverages a prospective cohort design with a large sample size, long follow-up duration, and comprehensive covariate assessment, enhancing the credibility and generalizability of the findings. Multiple validated dietary indices were applied, allowing for head-to-head comparison of dietary patterns within a unified analytical framework. The use of high-resolution geospatial air pollution data and rigorous sensitivity analyses further strengthens the reliability of the results. However, several limitations warrant consideration: First, as an observational study, our findings reflect associations rather than causality, and residual confounding or reverse causality cannot be fully excluded. Second, dietary intakes were self-reported, with ~40% of participants completing only one 24 h recall, introducing potential misclassification. Nonetheless, sensitivity analyses among those with ≥2 recalls showed consistent results. Third, air pollution exposure was estimated for 2010 only, potentially misclassifying long-term exposure, especially among participants who changed residence. Fourth, although improvements in the AUC, NRIs, and IDIs were statistically significant, their small magnitudes may limit their clinical utility. These metrics are more informative for population-level prevention than individual risk prediction. Finally, the predominantly White, UK-based population may limit generalizability to other populations, in which dietary habits, environmental exposures, and underlying disease susceptibility may differ. Future studies in more diverse cohorts are warranted to validate the observed associations.

## 5. Conclusions

Adherence to healthy dietary patterns is associated with a lower risk of COPD and may attenuate the adverse respiratory effects of ambient air pollution. While causality cannot be established, these findings support integrating nutrition into public health strategies for environmentally mediated respiratory disease.

## Figures and Tables

**Figure 1 nutrients-17-02029-f001:**
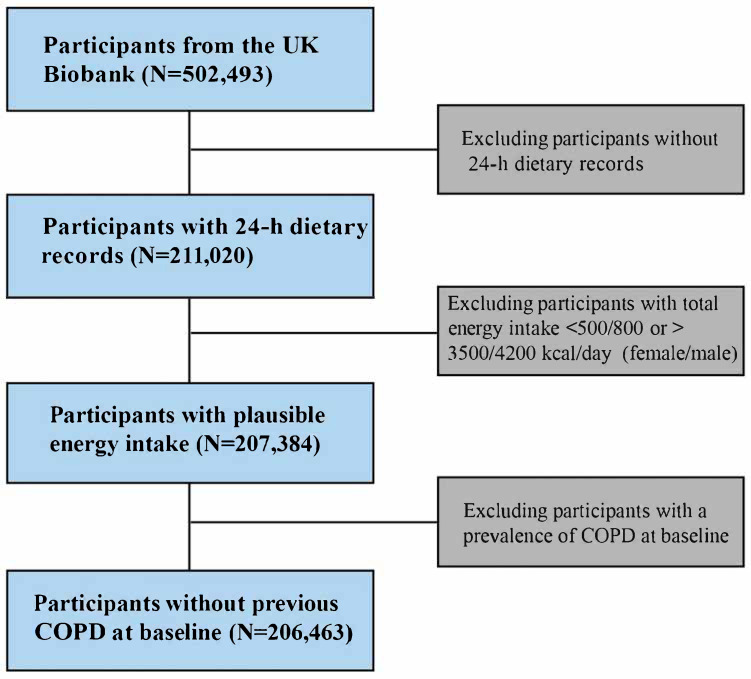
Flowchart of the study design based on the UK Biobank. Abbreviation: COPD, chronic obstructive pulmonary disease.

**Figure 2 nutrients-17-02029-f002:**
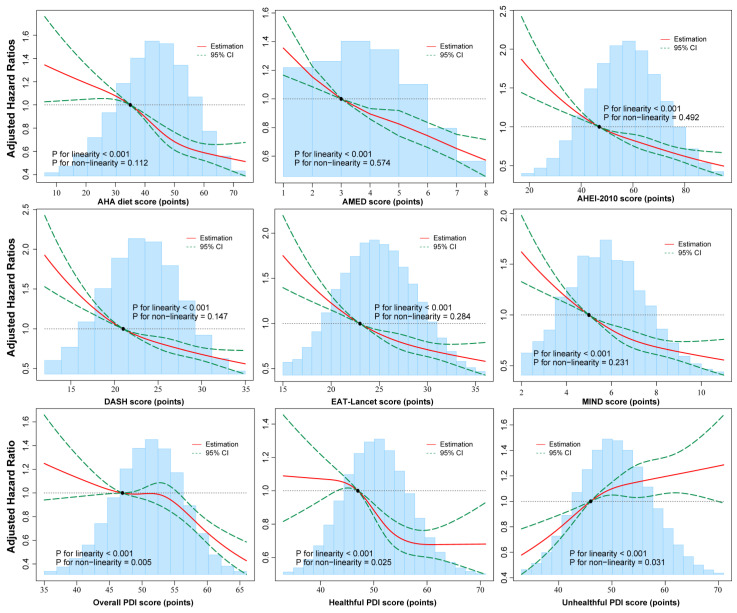
Multivariable-adjusted associations between dietary pattern scores and COPD incidence using restricted cubic spline regression. The reference point is the 25th percentile of the reference group from the categorical analysis with 4 knots. All models were adjusted for age, sex, ethnicity, Townsend index, education, BMI, physical activity, smoking, alcohol, total energy intake, CVD, T2DM, hypertension, other respiratory diseases, occupation-related respiratory problems, and all air pollution factors. Abbreviations: BMI, body mass index; CVD, cardiovascular disease; T2DM, type 2 diabetes mellitus.

**Figure 3 nutrients-17-02029-f003:**
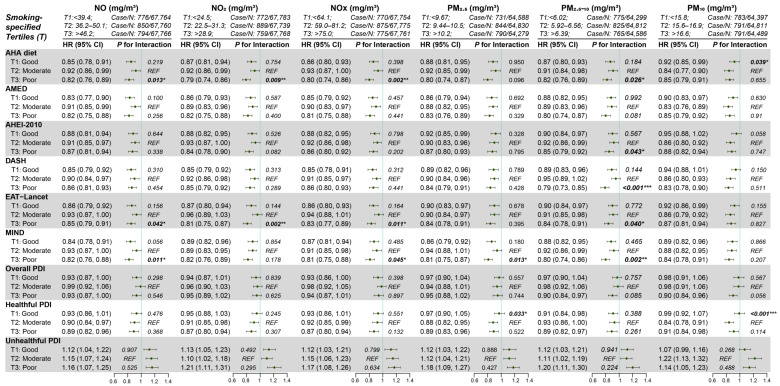
Hazard ratios (HR) and 95% confidence intervals (CI) for stratified analyses and modified effects by residential air pollution factors. All models were adjusted for age, sex, ethnicity, Townsend index, education, BMI, physical activity, smoking, alcohol, total energy intake, CVD, T2DM, hypertension, other respiratory diseases, and occupation-related respiratory problems. *, *p* < 0.05; **, *p* < 0.01; ***, *p* < 0.001.

**Table 1 nutrients-17-02029-t001:** Demographic characteristics across FEV1/FVC ratio quartiles at baseline (*n* = 206,463).

FEV1/FVC Ratio	Quartile 1 (<0.73)	Quartile 2 (0.73–0.77)	Quartile 3 (0.77–0.80)	Quartile 4 (>0.80)	Missing Data
	*n* = 38,499	*n* = 38,486	*n* = 38,229	*n* = 38,757	*n* = 52,492
FEV1 (L)	2.59 (2.12, 3.15)	2.82 (2.37, 3.39)	2.90 (2.46, 3.47)	3.01 (2.56, 3.60)	--
FVC (L)	3.81 (3.15, 4.61)	3.76 (3.16, 4.53)	3.70 (3.14, 4.43)	3.65 (3.11, 4.37)	--
Used inhaler within last hour (%)	472 (1.23)	230 (0.60)	165 (0.43)	124 (0.32)	204 (0.53)
Age (years)	60 (53, 64)	58 (51, 63)	56 (50, 62)	54 (47, 60)	57 (49, 63)
Male (%)	20,253 (52.61)	17,320 (45.00)	16,046 (41.97)	16,399 (42.31)	22,573 (43.00)
White race (%)	38,499 (100.00)	38,486 (100.00)	38,229 (100.00)	38,757 (100.00)	43,275 (82.44)
BMI (kg/m^2^)	25.6 (23.3, 28.5)	26.0 (23.6, 28.8)	26.4 (23.9, 29.4)	27.1 (24.4, 30.3)	26.3 (23.6, 29.5)
Smoking status (%)					
Never	18,483 (48.01)	21,177 (55.03)	22,200 (58.07)	23,809 (61.43)	31,197 (59.43)
Previous	15,496 (40.25)	14,407 (37.43)	13,557 (35.46)	12,756 (32.91)	16,884 (32.16)
Current	4432 (11.51)	2835 (7.37)	2402 (6.28)	2110 (5.44)	4177 (7.96)
Missing data	88 (0.23)	67 (0.17)	70 (0.18)	82 (0.21)	234 (0.45)
Alcohol consumption (%)					
Never	885 (2.30)	904 (2.35)	910 (2.38)	1018 (2.63)	2892 (5.51)
Previous	1167 (3.03)	1007 (2.62)	1033 (2.70)	1097 (2.83)	1872 (3.57)
Current	36,430 (94.63)	36,558 (94.99)	36,279 (94.90)	36,630 (94.51)	47,588 (90.66)
Missing data (%)	17 (0.04)	17 (0.04)	7 (0.02)	12 (0.03)	140 (0.27)
Physical activity (%)					
Low	5678 (14.75)	5609 (14.57)	5772 (15.10)	6384 (16.47)	8751 (16.67)
Moderate	13,724 (35.65)	13,891 (36.09)	13,948 (36.94)	14,058 (36.27)	18,585 (35.41)
High	13,426 (34.87)	13,328 (34.63)	12,930 (33.82)	12,633 (32.60)	16,324 (31.10)
Missing data (%)	5671 (14.73)	5658 (14.70)	5579 (14.59)	5682 (14.66)	8832 (16.83)
Total energy intake (kcal/day)	2078 (1737, 2472)	2047 (1715, 2431)	2022 (1693, 2397)	2008 (1678, 2385)	2006 (1661, 2406)
Townsend index	−2.36 (−3.75, −0.02)	−2.45 (−3.81, −0.23)	−2.45 (−3.78, −0.26)	−2.38 (−3.77, −0.12)	−2.01 (−3.57, 0.64)
Educational level (%)					
High	16,100 (41.8)	16,495 (42.9)	16,419 (43.0)	16,818 (43.4)	22,372 (42.6)
Moderate	12,608 (32.8)	13,249 (34.4)	13,226 (34.6)	13,638 (35.2)	16,883 (32.2)
Low	9791 (25.4)	8742 (22.7)	8584 (22.5)	8301 (21.4)	13,237 (25.2)
CVD (%)	2466 (6.41)	1972 (5.12)	1694 (4.43)	1575 (4.06)	3493 (6.65)
T2DM (%)	1242 (3.23)	1242 (3.23)	1264 (3.31)	1359 (3.51)	2559 (4.88)
Hypertension (%)	20,816 (54.1)	19,702 (51.2)	19,061 (49.9)	19,173 (49.5)	27,673 (52.7)
Respiratory diseases (%)	8009 (20.8)	4224 (11.0)	3186 (8.33)	2800 (7.22)	5867 (11.2)
Occupation-related breathing problems (%)	12,354 (32.1)	12,413 (32.3)	12,117 (31.7)	12,086 (31.2)	14,376 (27.4)
Residential air pollution (mg/m^3^)					
NO	41.6 (33.4, 49.9)	41.1 (33.1, 49.3)	41.1 (33.1, 49.2)	41.3 (33.5, 49.4)	42.9 (33.8, 51.3)
NO_2_	25.8 (20.7, 31.1)	25.6 (20.7, 30.7)	25.4 (20.7, 30.7)	25.6 (20.8, 30.8)	26.9 (21.8, 32.1)
NO_x_	67.5 (54.6, 80.5)	66.8 (54.3, 79.8)	66.7 (54.2, 79.6)	67.0 (54.6, 79.9)	69.9 (57.0, 83.0)
PM_2.5_	9.86 (9.21, 10.5)	9.83 (9.19, 10.5)	9.83 (9.19, 10.4)	9.86 (9.22, 10.5)	9.93 (9.31, 10.5)
PM_2.5–10_	6.11 (5.84, 6.61)	6.10 (5.84, 6.60)	6.11 (5.84, 6.61)	6.11 (5.84, 6.62)	6.16 (5.86, 6.68)
PM_10_	16.0 (15.2, 17.0)	16.0 (15.2, 17.0)	16.0 (15.2, 17.0)	16.0 (15.2, 17.0)	16.1 (15.4, 17.1)

Continuous variables are presented as medians with interquartile ranges, and categorical variables are expressed as numbers with corresponding percentages. Abbreviations: BMI, body mass index; CVD, cardiovascular disease; PM, particulate matter; T2DM, type 2 diabetes mellitus; NO, nitrogen oxide; NO_x_, nitrogen oxides; NO_2_, nitrogen dioxide.

**Table 2 nutrients-17-02029-t002:** Hazard ratios (HR) and 95% confidence intervals (CI) for dietary pattern scores in relation to chronic obstructive pulmonary disease risk.

	Case/*n*	Model 1		Model 2		Model 3	
		HR (95% CI)	*p* Value	HR (95% CI)	*p* Value	HR (95% CI)	*p* Value
AHA diet score							
Quintile 1	669/42,307	1 (Ref.)	<0.001	1 (Ref.)	<0.001	1 (Ref.)	<0.001
Quintile 2	588/42,388	0.87 (0.78, 0.98)		0.8 (0.72, 0.9)		0.94 (0.84, 1.05)	
Quintile 3	399/38,401	0.65 (0.58, 0.74)		0.57 (0.5, 0.65)		0.74 (0.65, 0.84)	
Quintile 4	419/40,712	0.65 (0.57, 0.73)		0.55 (0.49, 0.62)		0.76 (0.66, 0.86)	
Quintile 5	375/42,655	0.55 (0.49, 0.63)		0.44 (0.39, 0.5)		0.67 (0.58, 0.77)	
Z-score	2450/206,463	0.81 (0.78, 0.85)	<0.001	0.74 (0.71, 0.77)	<0.001	0.88 (0.84, 0.91)	<0.001
AMED score							
Quintile 1	565/36,727	1 (Ref.)	<0.001	1 (Ref.)	<0.001	1 (Ref.)	<0.001
Quintile 2	455/36,541	0.92 (0.82, 1.05)		0.87 (0.77, 0.99)		0.86 (0.76, 0.98)	
Quintile 3	525/42,919	0.80 (0.71, 0.90)		0.73 (0.65, 0.83)		0.84 (0.74, 0.95)	
Quintile 4	466/40,233	0.75 (0.67, 0.85)		0.68 (0.60, 0.76)		0.82 (0.72, 0.93)	
Quintile 5	439/40,233	0.69 (0.60, 0.79)		0.60 (0.53, 0.69)		0.66 (0.58, 0.76)	
Z-score	2450/206,463	0.83 (0.80, 0.87)	<0.001	0.76 (0.73, 0.79)	<0.001	0.87 (0.83, 0.91)	<0.001
AHEI-2010 score							
Quintile 1	655/40,483	1 (Ref.)	<0.001	1 (Ref.)	<0.001	1 (Ref.)	<0.001
Quintile 2	523/44,079	0.73 (0.65, 0.82)		0.75 (0.66, 0.84)		0.82 (0.73, 0.92)	
Quintile 3	464/39,434	0.73 (0.64, 0.82)		0.69 (0.62, 0.78)		0.81 (0.72, 0.91)	
Quintile 4	450/41,873	0.66 (0.59, 0.75)		0.63 (0.56, 0.72)		0.80 (0.70, 0.90)	
Quintile 5	358/40,594	0.54 (0.48, 0.62)		0.45 (0.4, 0.51)		0.70 (0.61, 0.79)	
Z-score	2450/206,463	0.81 (0.78, 0.84)	<0.001	0.75 (0.72, 0.78)	<0.001	0.90 (0.86, 0.93)	<0.001
DASH score							
Quintile 1	755/46,191	1 (Ref.)	<0.001	1 (Ref.)	<0.001	1 (Ref.)	<0.001
Quintile 2	365/29,724	0.75 (0.66, 0.85)		0.66 (0.58, 0.75)		0.82 (0.72, 0.93)	
Quintile 3	606/52,116	0.71 (0.64, 0.79)		0.58 (0.52, 0.65)		0.82 (0.73, 0.91)	
Quintile 4	317/31,590	0.61 (0.54, 0.7)		0.49 (0.43, 0.55)		0.75 (0.66, 0.86)	
Quintile 5	407/46,842	0.53 (0.47, 0.6)		0.4 (0.35, 0.45)		0.69 (0.61, 0.79)	
Z-score	2450/206,463	0.8 (0.77, 0.83)	<0.001	0.71 (0.68, 0.74)	<0.001	0.88 (0.85, 0.92)	<0.001
EAT-Lancet score							
Quintile 1	712/44,399	1 (Ref.)	<0.001	1 (Ref.)	<0.001	1 (Ref.)	<0.001
Quintile 2	463/35,129	0.82 (0.73, 0.92)		0.76 (0.68, 0.85)		0.9 (0.8, 1.01)	
Quintile 3	443/39,723	0.69 (0.62, 0.78)		0.62 (0.55, 0.69)		0.8 (0.71, 0.91)	
Quintile 4	532/50,390	0.66 (0.59, 0.73)		0.57 (0.51, 0.64)		0.81 (0.73, 0.91)	
Quintile 5	300/36,822	0.51 (0.44, 0.58)		0.43 (0.38, 0.5)		0.68 (0.60, 0.79)	
Z-score	2450/206,463	0.78 (0.75, 0.81)	<0.001	0.75 (0.72, 0.78)	<0.001	0.89 (0.85, 0.93)	<0.001
MIND score							
Quintile 1	700/40,525	1 (Ref.)	<0.001	1 (Ref.)	<0.001	1 (Ref.)	<0.001
Quintile 2	533/40,865	0.75 (0.67, 0.84)		0.70 (0.63, 0.79)		0.88 (0.78, 0.98)	
Quintile 3	500/46,329	0.62 (0.55, 0.7)		0.56 (0.5, 0.62)		0.76 (0.68, 0.85)	
Quintile 4	353/38,613	0.53 (0.46, 0.6)		0.46 (0.41, 0.53)		0.70 (0.61, 0.8)	
Quintile 5	364/40,131	0.52 (0.46, 0.59)		0.44 (0.38, 0.5)		0.72 (0.63, 0.82)	
Z-score	2450/206,463	0.75 (0.72, 0.78)	<0.001	0.72 (0.7, 0.76)	<0.001	0.88 (0.84, 0.91)	<0.001
Overall PDI score							
Quintile 1	522/41,525	1 (Ref.)	<0.001	1 (Ref.)	<0.001	1 (Ref.)	<0.001
Quintile 2	474/37,372	1.01 (0.89, 1.14)		0.95 (0.84, 1.08)		1.08 (0.95, 1.22)	
Quintile 3	544/44,877	0.96 (0.85, 1.09)		0.87 (0.77, 0.98)		1.04 (0.92, 1.18)	
Quintile 4	508/40,539	1 (0.88, 1.12)		0.88 (0.78, 0.99)		1.10 (0.97, 1.24)	
Quintile 5	402/42,150	0.76 (0.66, 0.86)		0.65 (0.57, 0.74)		0.86 (0.75, 0.98)	
Z-score	2450/206,463	0.92 (0.88, 0.95)	<0.001	0.87 (0.83, 0.9)	<0.001	0.95 (0.92, 0.99)	0.025
Healthful PDI score							
Quintile 1	574/41,574	1 (Ref.)	<0.001	1 (Ref.)	<0.001	1 (Ref.)	<0.001
Quintile 2	601/44,376	0.98 (0.87, 1.1)		0.89 (0.8, 1)		1.00 (0.89, 1.12)	
Quintile 3	465/37,624	0.89 (0.79, 1.01)		0.78 (0.69, 0.89)		0.97 (0.86, 1.10)	
Quintile 4	407/42,166	0.7 (0.61, 0.79)		0.59 (0.52, 0.67)		0.75 (0.65, 0.85)	
Quintile 5	403/40,723	0.71 (0.63, 0.81)		0.6 (0.53, 0.68)		0.82 (0.72, 0.94)	
Z-score	2450/206,463	0.87 (0.84, 0.91)	<0.001	0.81 (0.78, 0.85)	<0.001	0.91 (0.87, 0.95)	<0.001
Unhealthful PDI score							
Quintile 1	408/40,958	1 (Ref.)	<0.001	1 (Ref.)	<0.001	1 (Ref.)	<0.001
Quintile 2	511/46,503	1.1 (0.97, 1.26)		1.2 (1.05, 1.36)		1.19 (1.04, 1.36)	
Quintile 3	449/34,414	1.31 (1.15, 1.5)		1.44 (1.26, 1.65)		1.38 (1.21, 1.59)	
Quintile 4	572/43,973	1.31 (1.15, 1.48)		1.54 (1.36, 1.75)		1.39 (1.22, 1.59)	
Quintile 5	510/40,615	1.26 (1.11, 1.44)		1.59 (1.4, 1.81)		1.34 (1.16, 1.54)	
Z-score	2450/206,463	1.1 (1.06, 1.15)	<0.001	1.21 (1.16, 1.26)	<0.001	1.13 (1.08, 1.18)	<0.001

Quintiles for each dietary pattern score are presented as the merging of sex-specific quintiles. Model 1 is the unadjusted model. Model 2 was adjusted for age, sex, and ethnicity. Model 3 includes further adjustments for Townsend deprivation scores, educational attainment, physical activity levels, smoking status, alcohol consumption, total energy intake, BMI, CVD, hypertension, T2DM, respiratory diseases, occupation-related breathing problems, and air pollution factors including, NO, NO_2_, PM_2.5_, PM_2.5–10_, and PM_10_. Abbreviations: CVD, cardiovascular disease; PM, particulate matter; T2DM, type 2 diabetes mellitus; NO, nitrogen oxides; NO_2_, nitrogen dioxide.

## Data Availability

Researchers registered with UK Biobank can apply for access to the database by completing an application (https://www.ukbiobank.ac.uk/enable-your-research/apply-for-access, accessed on 8 July 2021). The code supporting this study is available upon reasonable request to the corresponding author at lhchen@ntu.edu.cn, because of restrictions on data sharing imposed by the UK Biobank and the context-specific nature of the analysis code, which is tailored to the licensed dataset.

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
