# Peer review of "Dietary Habits, Residential Air Pollution, and Chronic Obstructive Pulmonary Disease"

_nutrients, 2025, doi:10.3390/nu17122029_

Round 1
Reviewer 1 Report
Comments and Suggestions for Authors
Reviewer Report
Title: Dietary Habits, Residential Air Pollution and Chronic Obstructive Pulmonary Disease
Journal: Nutrients
General Assessment
This manuscript investigates the association between adherence to multiple dietary patterns and the risk of incident COPD, including potential effect modification by residential air pollution. The study addresses a relevant public health question and leverages a large, well-characterized dataset from the UK Biobank. The methodological framework is robust, and the analyses are extensive. However, there are several notable limitations that should be addressed to improve the scientific rigor and interpretability of the findings.
Major Weaknesses
- The study design is observational, and although multiple covariates are adjusted for, residual confounding and reverse causality cannot be ruled out. The authors should more clearly state that causality cannot be inferred and temper their conclusions accordingly.
- Air pollution exposure is modeled for a single year (2010), which may not accurately reflect long-term exposure, especially given potential changes in residential address or pollution levels over time. The authors should discuss this limitation more explicitly.
- The reliance on 24-hour dietary recall data (with only a subset of participants having repeated measures) introduces potential for dietary misclassification. Clarification is needed on how many recalls were available per participant and how this may affect reliability.
- The study population is predominantly White and UK-based, which restricts the applicability of the findings to more diverse global populations. The authors should acknowledge this limitation and consider discussing implications for other demographic groups.
- While most healthy dietary patterns were protective, the Overall and Healthful PDI scores showed weaker or inconsistent associations. The manuscript would benefit from a more detailed interpretation of these discrepancies.
- COPD incidence is partially based on self-reported data, which may introduce bias or misclassification. The extent of reliance on self-report versus clinically verified outcomes should be clarified.
- Although statistically significant, the improvements in AUC, NRI, and IDI are numerically very small and may lack clinical relevance. The discussion should include a more critical evaluation of these results.
- The hypothesized biological mechanisms (e.g., oxidative stress modulation) are not substantiated with biomarker or intermediate phenotype data. This weakens the mechanistic plausibility of the observed associations.
Minor Issues
- Clarify whether the baseline spirometry used FEV₁/FVC pre- or post-bronchodilator values.
- Consider simplifying the presentation of dietary indices in the main text to improve reader comprehension (too many acronyms and overlapping constructs).
- Some grammar and phrasing errors exist (e.g., "a little study evaluated" should be "few studies have evaluated").
Author Response
General Assessment
This manuscript investigates the association between adherence to multiple dietary patterns and the risk of incident COPD, including potential effect modification by residential air pollution. The study addresses a relevant public health question and leverages a large, well-characterized dataset from the UK Biobank. The methodological framework is robust, and the analyses are extensive. However, there are several notable limitations that should be addressed to improve the scientific rigor and interpretability of the findings.
Response: We sincerely thank the reviewer for their insightful feedback, which has strengthened our manuscript. Below are point-by-point responses to each concern, with corresponding revisions highlighted in the manuscript (tracked changes).
Major Weaknesses
The study design is observational, and although multiple covariates are adjusted for, residual confounding and reverse causality cannot be ruled out. The authors should more clearly state that causality cannot be inferred and temper their conclusions accordingly.
Response: We sincerely thank the reviewer for highlighting this crucial limitation. Notwithstanding these limitations, large prospective cohorts like UK Biobank provide unique advantages for studying complex diet-environment-disease relationships. Randomized trials of long-term dietary interventions in high-pollution settings are impractical, making well-controlled observational evidence critical for generating hypotheses.
We fully acknowledge that observational designs preclude causal inference. We have implemented the following revisions to ensure balanced interpretation:
Abstract: "Adherence to AHA, EAT-Lancet, and MIND diets is associated with reduced COPD risk, with potentially amplified benefits in high-pollution areas.
Results: Notably, compared to the moderate exposures (the second tertile, T2), the apparent protective associations were most pronounced of AHA diet in poor air condition
Discussion: “However, several limitations warrant consideration. First, as an observational study, our findings reflect associations rather than causality, and residual confounding or reverse causality cannot be fully excluded.”
Conclusion: “Adherence to healthy dietary patterns is associated with a lower risk of COPD and may attenuate the adverse respiratory effects of ambient air pollution. While causality cannot be established, these findings support integrating nutrition into public health strategies for environmentally mediated respiratory disease.”
Air pollution exposure is modeled for a single year (2010), which may not accurately reflect long-term exposure, especially given potential changes in residential address or pollution levels over time. The authors should discuss this limitation more explicitly.
Response: Thank you for your valuable comment. We fully acknowledge that estimating air pollution exposure based on a single year (2010) may not fully capture long-term exposure patterns, particularly given potential changes in residential addresses and temporal variations in pollution levels. Due to data limitations, we were unable to assess multi-year air pollution exposure. To address this important limitation, we have now explicitly discussed this point in the revised manuscript:
Limitations: “Third, air pollution exposure was estimated for 2010 only, potentially misclassifying long-term exposure, especially among participants who changed residence.”
The reliance on 24-hour dietary recall data (with only a subset of participants having repeated measures) introduces potential for dietary misclassification. Clarification is needed on how many recalls were available per participant and how this may affect reliability.
Response: Thank you for this insightful comment. We acknowledge that the use of 24-hour dietary recall data may introduce measurement error and potential dietary misclassification, particularly due to day-to-day variability in individual dietary intake. As such, the single-day recall may not fully represent the usual intake at the individual level. UK Biobank's online 24hr recall tool shows moderate-to-good reliability for food groups. Previous validation studies in large epidemiological cohorts, including the UK Biobank, have shown that even single 24-hour recalls can reasonably characterize average dietary patterns at the population level and are suitable for ranking individuals in prospective analyses [1]. Furthermore, in our study, over half of participants had repeated dietary recalls on non-consecutive days (61%, n= 125319).
We have now added further clarification on the number of recalls per participant and discussed the potential impact on reliability and misclassification bias in the revised manuscript:
Material & Methods: “A subset of participants (61%, n= 125319) completed repeated 24-hour dietary recalls during follow-up: 47,403 participants completed two assessments, 42,013 completed three, 30,192 completed four, and 5,711 completed five in the present study.”
Discussion-Limitation: “Second, dietary intakes were self-reported, with ~40% of participants completing only one 24-hour recall, introducing potential misclassification. Nonetheless, sensitivity analyses among those with ≥2 recalls showed consistent results.”
The study population is predominantly White and UK-based, which restricts the applicability of the findings to more diverse global populations. The authors should acknowledge this limitation and consider discussing implications for other demographic groups.
Response: Thank you for highlighting this important point. We agree that the generalizability of our findings may be limited due to the demographic composition of the UK Biobank cohort, which is predominantly White and UK-based. This limitation may restrict the applicability of our results to more ethnically and geographically diverse populations, where dietary habits, environmental exposures, and genetic susceptibility may differ.
We have now explicitly acknowledged this limitation in the revised Discussion section: “Finally, the predominantly White, UK-based population may limit generalizability to other populations, in which dietary habits, environmental exposures, and underlying disease susceptibility may differ. Future studies in more diverse cohorts are warranted to validate the observed associations.”
While most healthy dietary patterns were protective, the Overall and Healthful PDI scores showed weaker or inconsistent associations. The manuscript would benefit from a more detailed interpretation of these discrepancies.
Response: Thank you for this thoughtful comment. We appreciate your observation regarding the weaker or inconsistent associations observed for the Overall Plant-Based Diet Index (PDI) and Healthful PDI, compared with other dietary indices. In the revised manuscript, we have added further interpretation to explain these discrepancies.
“While most healthy dietary patterns were inversely associated with COPD risk, the Overall PDI showed inconsistent associations. This may be due to its construction, which assigns equal positive scores to all plant foods regardless of nutritional quality, including refined grains, sugar-sweetened beverages, and other less healthful items. Similarly, although the Healthful PDI emphasizes nutritious plant-based foods, it also penalizes all animal-based foods including fish and dairy—which may have beneficial effects on respiratory health [2]. These scoring characteristics may partly explain the attenuated associations observed. Furthermore, Healthful PDI also demonstrated weaker associations in participants exposed to lower levels of air pollution. This could reflect a lower baseline risk of COPD, a reduced oxidative burden, and a potential ceiling effect of dietary benefits in cleaner environments, where the protective role of diet may be less pronounced.”
COPD incidence is partially based on self-reported data, which may introduce bias or misclassification. The extent of reliance on self-report versus clinically verified outcomes should be clarified.
Thank you for this important comment. We fully agree that reliance on self-reported data can introduce potential misclassification and bias in disease outcome assessment. To address this, we carefully defined incident COPD using multiple sources of information available in the UK Biobank. Specifically, COPD cases (n = 2,450) were identified through a combination of:
- Linked hospital inpatient records and death registry data, coded using ICD-9 (491.2, 496) and ICD-10 (J44), and
- Self-reported physician-diagnosed COPD, obtained during baseline and follow-up assessments.
To assess the extent of overlap and potential bias, we cross-tabulated COPD ascertainment by ICD codes and self-reported diagnosis:
|
|
|
Self-report COPD |
|
|
|
|
Non-COPD |
COPD |
|
ICD-9/10 |
Non-COPD |
204013 |
5 |
|
COPD |
2444 |
1 |
|
We have now clarified this point in the revised Results: “A total of 2,450 incident COPD cases were identified during follow-up. The vast majority of cases were ascertained through linked clinical records (hospital inpatient or death registry), with only a negligible number (n = 5) based solely on self-reported diagnoses without supporting clinical documentation.”
In addition, we conducted a sensitivity analysis excluding cases identified only through self-report (n = 5). The associations between dietary patterns and COPD risk remained materially unchanged, demonstrating the robustness of our findings:
|
Case/N |
2445/206458 |
|
|
Dietary Pattern Score |
HR (95%CI) |
P value |
|
AHA diet score |
0.87 (0.83, 0.90) |
<0.001 |
|
AMED score |
0.86 (0.82, 0.90) |
<0.001 |
|
AHEI score |
0.89 (0.85, 0.92) |
<0.001 |
|
DASH score |
0.87 (0.84, 0.91) |
<0.001 |
|
EAT-Lancet score |
0.88 (0.84, 0.92) |
<0.001 |
|
MIND score |
0.87 (0.83, 0.90) |
<0.001 |
|
Overall PDI score |
0.95 (0.91, 0.99) |
0.020 |
|
Healthful PDI score |
0.91 (0.87, 0.95) |
<0.001 |
|
Unhealthful PDI score |
1.14 (1.09, 1.19) |
<0.001 |
Although statistically significant, the improvements in AUC, NRI, and IDI are numerically very small and may lack clinical relevance. The discussion should include a more critical evaluation of these results.
Response: Thank you for this insightful comment. Although the improvements in discrimination metrics such as AUC, NRI, and IDI were statistically significant, their absolute magnitudes were modest and may have limited clinical utility.
Specifically, we now acknowledge that although the inclusion of dietary indices modestly improved model performance for predicting COPD risk, the observed gains in AUC, NRI, and IDI are unlikely to result in meaningful improvements in individual risk prediction or clinical decision-making. This is a common challenge in epidemiological modeling, particularly for exposures such as dietary patterns, where even well-established dietary factors may only yield modest enhancements in predictive discrimination [3]. We now emphasize that the principal value of our findings lies in identifying diet as a modifiable risk factor for COPD at the population level, rather than in its immediate applicability for clinical risk stratification. A corresponding statement has been added to the Discussion section of the manuscript:
Limitations: "Fourth, although improvements in AUC, NRI, and IDI were statistically significant, their small magnitudes may limit clinical utility. These metrics are more informative for population-level prevention than individual risk prediction."
The hypothesized biological mechanisms (e.g., oxidative stress modulation) are not substantiated with biomarker or intermediate phenotype data. This weakens the mechanistic plausibility of the observed associations.
Response: Thank you for this valuable comment. We fully acknowledge that our study did not include biomarker or intermediate phenotype data (e.g., markers of oxidative stress, inflammation, or lung function trajectories), which limits our ability to directly substantiate the proposed biological mechanisms. The hypothesized pathways—such as dietary modulation of oxidative stress, systemic inflammation, and immune responses—were based on established evidence from prior mechanistic and clinical studies. These studies have shown that antioxidant-rich diets, such as those high in fruits, vegetables, whole grains, and healthy fats, may attenuate oxidative damage and inflammatory responses implicated in the pathogenesis of COPD [4,5].
To address this limitation, we have now revised the Discussion section to explicitly acknowledge that while our findings are biologically plausible, they remain observational and indirect in nature. We emphasize the need for future studies incorporating longitudinal biomarker data or functional respiratory outcomes to validate the proposed mechanisms.
“Although the inverse associations between healthy dietary patterns and COPD risk are biologically plausible—primarily due to their potential anti-inflammatory and antioxidant effects [4]. our study lacked biomarker data (e.g., CRP, IL-6, oxidative stress markers) or intermediate phenotypes to directly substantiate these mechanistic pathways. Therefore, further longitudinal and mechanistic studies incorporating such data are warranted.”
Minor Issues
Clarify whether the baseline spirometry used FEV₁/FVC pre- or post-bronchodilator values.
Response: Thank you for this important comment. According to the UK Biobank protocol, spirometry at baseline was conducted without the administration of bronchodilators. Before testing, participants were screened for contraindications (e.g., recent chest infection, recent surgeries, pregnancy), and spirometry was not performed if any were present. Although no bronchodilators were administered as part of the assessment, it was recorded whether participants had used an inhaler within the hour preceding the test.
In our analysis, FEV₁ and FVC values were derived from all available spirometry data, with only a small proportion of participants (~0.62%) reporting inhaler use before the test. Therefore, our primary analysis included all participants with valid lung function measurements, regardless of recent inhaler use. As shown in Table 1, individuals in higher FEV₁/FVC ratio quartiles tended to be younger, more likely female, with lower prevalence of current smoking, reduced total energy intake, fewer cardiopulmonary comorbidities, and less frequent inhaler use before the spirometry test.
|
FEV1/FVC ratio |
Quartile 1 (<0.73) |
Quartile 2 (0.73-0.77) |
Quartile 3 (0.77-0.80) |
Quartile 4 (>0.80) |
Missing data |
|
N=38499 |
N=38486 |
N=38229 |
N=38757 |
N=52492 |
|
|
FEV1 (L) |
2.59 (2.12, 3.15) |
2.82 (2.37, 3.39) |
2.90 (2.46, 3.47) |
3.01 (2.56, 3.60) |
-- |
|
FVC (L) |
3.81 (3.15, 4.61) |
3.76 (3.16, 4.53) |
3.70 (3.14, 4.43) |
3.65 (3.11, 4.37) |
-- |
|
Used inhaler within last hour (%) |
472 (1.23) |
230 (0.60) |
165 (0.43) |
124 (0.32) |
204 (0.53) |
We have added this clarification to the Methods and Results sections of the revised manuscript:
Material & Methods: “The procedure included checks for contraindications (e.g., recent chest infection, recent surgery, pregnancy), and participants reporting any were excluded from testing. Furthermore, it was recorded whether participants had used an inhaler within the last hour.”
Results: “Notably, individuals in higher FEV₁/FVC ratio quartiles tended to be younger, more likely female, and had healthier profiles overall—including lower rates of current smoking, reduced total energy intake, fewer cardiovascular comorbidities (e.g., CVD and hypertension), fewer respiratory disease diagnoses, and less frequent inhaler use within the hour prior to spirometry testing.”
Consider simplifying the presentation of dietary indices in the main text to improve reader comprehension (too many acronyms and overlapping constructs).
Response:
Thank you for this helpful suggestion. We agree that presenting multiple dietary indices with overlapping components and numerous acronyms may hinder clarity for readers. To address this, we have revised the main text to simplify the presentation of dietary indices.
Specifically:
“Figure 2 and e-Figure 2 present multivariable-adjusted restricted cubic spline analyses revealing significant linear associations for general healthy dietary patterns with both risks of COPD (all P for linearity < 0.001) and COPD-related mortality (all P for linearity < 0.05).”
Some grammar and phrasing errors exist (e.g., "a little study evaluated" should be "few studies have evaluated").
Response: Thank you for your careful reading and valuable comment. We have thoroughly reviewed the manuscript for grammatical and phrasing errors and made appropriate revisions to improve the clarity and fluency of the text.
References
- Bradbury, K.E.; Young, H.J.; Guo, W.; Key, T.J. Dietary assessment in UK Biobank: an evaluation of the performance of the touchscreen dietary questionnaire. J Nutr Sci 2018, 7, e6, doi:10.1017/jns.2017.66.
- Varraso, R.; Barr, R.G.; Willett, W.C.; Speizer, F.E.; Camargo, C.A. Fish intake and risk of chronic obstructive pulmonary disease in 2 large US cohorts. Am J Clin Nutr 2015, 101, 354-361, doi:10.3945/ajcn.114.094516.
- Lu, X.; Wu, L.; Shao, L.; Fan, Y.; Pei, Y.; Lu, X.; Borné, Y.; Ke, C. Adherence to the EAT-Lancet diet and incident depression and anxiety. Nature Communications 2024, 15, doi:10.1038/s41467-024-49653-8.
- Barnes, P.J. Inflammatory mechanisms in patients with chronic obstructive pulmonary disease. J Allergy Clin Immunol 2016, 138, 16-27, doi:10.1016/j.jaci.2016.05.011.
- Kelly, F.J.; Fussell, J.C. Air pollution and airway disease. Clin Exp Allergy 2011, 41, 1059-1071, doi:10.1111/j.1365-2222.2011.03776.x.
- Marchese, L.E.; McNaughton, S.A.; Hendrie, G.A.; Machado, P.P.; O’Sullivan, T.A.; Beilin, L.J.; Mori, T.A.; Dickinson, K.M.; Livingstone, K.M. Trajectories of plant-based dietary patterns and their sex-specific associations with cardiometabolic health among young Australian adults. International Journal of Behavioral Nutrition and Physical Activity 2025, 22, doi:10.1186/s12966-025-01765-0.
- Vienneau, D.; de Hoogh, K.; Bechle, M.J.; Beelen, R.; van Donkelaar, A.; Martin, R.V.; Millet, D.B.; Hoek, G.; Marshall, J.D. Western European Land Use Regression Incorporating Satellite- and Ground-Based Measurements of NO2 and PM10. Environmental Science & Technology 2013, 47, 13555-13564, doi:10.1021/es403089q.
Reviewer 2 Report
Comments and Suggestions for Authors
The idea of the study is interesting and valuable. Its results are potentially useful from a scientific and clinical practice perspective. The manuscript indicates that there are 3 figures. However, I did not find them in the manuscript. Maybe they contain information that I missed. I therefore put the following questions:
- In what year was the subjects' diet type confirmed?
- In what year was it confirmed that the subjects already had COPD?
- How was the duration of the diet determined in the individual case? Diet can only influence whether or not a disease, such as COPD, develops if it is used for a very long time.
- How was it known that the diet was not changed during the study period?
- How was the duration of air pollution assessed? How was the variation in air pollution assessed? Air pollution, like diet, is dynamic and depends on many factors. For example, changes in place of residence, changes in transport flows, workplace, etc.
I would like to get answers to these questions so that I can make a final assessment of the manuscript.
Author Response
Comments and Suggestions for Authors
The idea of the study is interesting and valuable. Its results are potentially useful from a scientific and clinical practice perspective. The manuscript indicates that there are 3 figures. However, I did not find them in the manuscript. Maybe they contain information that I missed. I therefore put the following questions:
Response: We sincerely apologize for the omission. We have now re-attached them in the revised manuscript and ensured they are correctly cited in the text. These figures include:
Figure 1: Flowchart of the study design based on the UK biobank.
Figure 2: Multivariable-adjusted associations between dietary pattern scores and COPD incidence using restricted cubic spline regression.
Figure 3: Hazard ratios (HR) and 95% confidence intervals (CI) for stratified analyses and modified effects by residential air pollution factors.
In what year was the subjects' diet type confirmed?
Response: The dietary data used in our study were collected during the baseline assessment of the UK Biobank, which was conducted between April 2009 and June 2012. At baseline, participants completed a touchscreen dietary questionnaire assessing habitual intake. In addition, a subset of participants completed repeated 24-hour dietary recall assessments during follow-up. This has now been clarified in the revised manuscript. The distribution of participants by the number of dietary recall assessments is provided below for reference:
Material & Methods: “Between April 2009 and June 2012, dietary intake was assessed using a validated online 24-hour dietary recall tool, developed specifically for large population-based studies [25]. A subset of participants completed repeated 24-hour dietary recalls during follow-up: 47,403 participants completed two assessments, 42,013 completed three, 30,192 completed four, and 5,711 completed five in the present study.”
In what year was it confirmed that the subjects already had COPD?
Response: In our study, the presence of COPD at baseline was determined using multiple data sources, including linked hospital inpatient records (based on ICD-9/10 codes), and self-reported physician diagnoses collected at the time of recruitment. Baseline assessments in the UK Biobank were conducted between April 2006 and December 2010. Participants were classified as having prevalent COPD at baseline if they either (1) self-reported a physician diagnosis of COPD during baseline assessment or (2) had a hospital inpatient record indicating a COPD diagnosis dated prior to their recruitment. These individuals were excluded from the incident COPD analysis to ensure the outcome reflected new-onset cases during follow-up.
How was the duration of the diet determined in the individual case? Diet can only influence whether or not a disease, such as COPD, develops if it is used for a very long time.
Response: Thank you for this important comment. We agree that the duration of dietary exposure is critical when assessing its relationship with chronic diseases such as COPD. In our study, dietary intake was assessed using the Oxford WebQ, a validated web-based 24-hour dietary recall questionnaire, which was first administered at baseline between 2009 and 2012. A subset of participants completed up to five repeated 24-hour dietary recalls over a period of time, allowing us to better capture habitual dietary intake for those individuals. Specifically, 47,403 participants completed two assessments, 42,013 completed three, 30,192 completed four, and 5,711 completed five. To estimate longer-term dietary habits, we averaged the dietary intakes across all available dietary recalls for each participant, which is a widely accepted method in nutritional epidemiology to reduce within-person variability and better reflect habitual intake.
How was it known that the diet was not changed during the study period?
Response: Thank you for your thoughtful question. In our study, dietary intake was primarily assessed at baseline using the Oxford WebQ 24-hour dietary recall tool. For a subset of participants, repeated 24-hour dietary recalls (up to five times) were conducted over a follow-up period to better capture habitual dietary intake. We used the average of all available dietary assessments to estimate each participant’s typical dietary pattern and reduce within-person variation.
However, as is common in large cohort studies, we did not have continuous or annual dietary tracking for all participants. Therefore, we cannot completely rule out the possibility of dietary changes over the follow-up period. Furthermore, previous studies have shown that dietary patterns tend to be relatively stable over the life stages [1], particularly over short-to-moderate time frames. Therefore, while we acknowledge the limitation of not having time-updated dietary data for the entire cohort, we believe that our use of repeated assessments (where available) and exclusion of prevalent cases help to ensure that the dietary exposures reflected pre-disease dietary habits.
How was the duration of air pollution assessed? How was the variation in air pollution assessed? Air pollution, like diet, is dynamic and depends on many factors. For example, changes in place of residence, changes in transport flows, workplace, etc.
Response: Thank you for this important comment. We agree that air pollution exposure is dynamic and influenced by various time- and location-dependent factors. In our study, long-term exposure to air pollution was assessed using participants’ residential addresses between 26 January 2010 and 18 January 2011, with the air pollution estimates representing average exposure for the year 2010 (https://biobank.ctsu.ox.ac.uk/crystal/label.cgi?id=114 ). Specifically, we employed land use regression (LUR) models developed as part of the European Study of Cohorts for Air Pollution Effects (ESCAPE) project to estimate annual average concentrations of several pollutants — including nitrogen oxide (NO), nitrogen dioxide (NO₂), particulate matter with aerodynamic diameters <2.5 μm (PM₂.₅), between 2.5–10 μm (PM₂.₅–₁₀), and <10 μm (PM₁₀). These LUR models provided high spatial resolution estimates based on traffic density, land use, population, and other geographic predictors. In addition, EU-wide air pollution maps were generated using enhanced LUR models incorporating satellite-derived air pollution estimates to further improve performance. Validation studies have demonstrated that these models reliably capture spatial variation in annual mean levels of NOx and PM [2].
To address potential variations in air pollution exposure due to residential or occupational mobility, we conducted sensitivity analyses by excluding participants who reported occupation-related respiratory problems or had diagnosed respiratory diseases at baseline. The consistency of results across these analyses suggests that our findings are robust despite potential exposure misclassification.
We acknowledge that our study does not capture temporal fluctuations in air pollution levels, nor does it account for changes in workplace environments, commuting behaviors, or local traffic dynamics over time. These limitations have now been explicitly discussed in the revised Discussion section of the manuscript.
“Third, air pollution exposure was estimated for 2010 only, potentially misclassifying long-term exposure, especially among participants who changed residence.”
I would like to get answers to these questions so that I can make a final assessment of the manuscript.
References
- Marchese, L.E.; McNaughton, S.A.; Hendrie, G.A.; Machado, P.P.; O’Sullivan, T.A.; Beilin, L.J.; Mori, T.A.; Dickinson, K.M.; Livingstone, K.M. Trajectories of plant-based dietary patterns and their sex-specific associations with cardiometabolic health among young Australian adults. International Journal of Behavioral Nutrition and Physical Activity 2025, 22, doi:10.1186/s12966-025-01765-0.
- Vienneau, D.; de Hoogh, K.; Bechle, M.J.; Beelen, R.; van Donkelaar, A.; Martin, R.V.; Millet, D.B.; Hoek, G.; Marshall, J.D. Western European Land Use Regression Incorporating Satellite- and Ground-Based Measurements of NO2 and PM10. Environmental Science & Technology 2013, 47, 13555-13564, doi:10.1021/es403089q.
Round 2
Reviewer 1 Report
Comments and Suggestions for Authors
none
Reviewer 2 Report
Comments and Suggestions for Authors
I thank the authors for considering my comments and correcting the manuscript accordingly.
I have no additional comments.